# Predicting the Outcome of Equine Artificial Inseminations Using Chilled Semen

**DOI:** 10.3390/ani13071203

**Published:** 2023-03-30

**Authors:** Ashlee Jade Medica, Sarah Lambourne, Robert John Aitken

**Affiliations:** 1Priority Research Centre for Reproductive Science, Discipline of Biological Sciences, School of Environmental and Life Sciences, College of Engineering Science and Environment, University of Newcastle, Callaghan, NSW 2308, Australia; 2Hunter Medical Research Institute, New Lambton Heights, NSW 2305, Australia

**Keywords:** stallion fertility, semen analysis, pregnancy prediction, artificial insemination, WST-1, samson™ system

## Abstract

**Simple Summary:**

The outcome of equine artificial inseminations can be predicted with a high level of accuracy—87.9% if analyzed pre-chilling, and 95% if analyzed post-chilling—by optimizing a fertility prediction algorithm for individual stallions in conjunction with the Samson™ system and the reduction of the probe WST-1.

**Abstract:**

This study aimed to determine whether an analysis of stallion ejaculate could accurately predict the likelihood of pregnancy resulting from artificial insemination in mares. This study involved 46 inseminations of 41 mares, using 7 standardbred stallions over a 5-week period at an Australian pacing stud. Semen quality was assessed immediately after collection and again after chilling at ~5 °C for 24 h. The assessment involved evaluating ejaculate volume, sperm concentration, and motility parameters using an iSperm^®^ Equine portable device. After the initial evaluation, a subpopulation of cells was subjected to a migration assay through a 5 µm polycarbonate filter within a Samson™ isolation chamber over a 15 min period. The cells were assessed for their concentration, motility parameters, and ability to reduce the membrane impermeant tetrazolium salt WST-1. The data, combined with the stallion and mare’s ages, were used to predict the likelihood of pregnancy, as confirmed by rectal ultrasound sonography performed 14 days post ovulation. The criteria used to predict pregnancy were optimized for each individual stallion, resulting in an overall accuracy of 87.9% if analyzed pre-chilling and 95% if analyzed post-chilling. This study suggests that an analysis of stallion ejaculate can be used to predict the likelihood of pregnancy resulting from artificial insemination in mares with a high level of accuracy.

## 1. Introduction

Standardbred horses, as with most domestic animals, are selected based on their athleticism and conformationally correct phenotype and not based on their fertility. These selection methods, although undeniably producing champions, have consequently led to a decline in fertility [1,2]. While limited information is available for the per cycle conception rates in standardbred horses alone, other breeds, which also utilize artificial insemination (AI), have indicated pregnancy rates per cycle range from 76% to 83% [3,4,5] after AI using fresh semen and pregnancy rates of 44–69% after AI using chilled semen [5,6,7,8]. Several factors have been observed to impact reproductive efficiency, including the age of the stallion and mare, the mare’s reproductive history (e.g., maiden, barren, foal heat), the month of breeding, genotype, frequency of mating, and endocrine levels [9,10,11,12,13].

The reproductive efficiency of horses has improved over the last three decades, primarily due to veterinary research and intervention [14]. Nevertheless, the equine breeding industry is still facing major problems due to the ongoing lack of fertility. As seasonal breeders, the equine breeding season runs from September–February in the Southern Hemisphere and from February–July in the Northern Hemisphere. With foals born earlier in the season, considered to have a strong competitive advantage, there is a clear drive to achieve pregnancies as early in the breeding season as possible. A method of determining whether insemination will result in a positive pregnancy at the time of breeding would be of immense benefit to the industry. If, in a given circumstance, pregnancy was believed unlikely, secondary insemination during the same estrus cycle could be performed. Such a strategy should greatly increase the chance of pregnancy, without awaiting 14 days for a negative rectal ultrasound examination and subsequent time for the mare to return to estrus, thus improving a stallion’s per cycle conception rate.

The Samson™ isolation chamber was developed by Memphasys Ltd. (Sydney, Australia) to isolate a subpopulation of highly motile cells. This is achieved by placing an ejaculate within the Samson™ device, where the spermatozoa are faced with a motility challenge (5 µm polycarbonate filter). Highly motile cells may locate and swim through the pores within the membrane into a separate chamber (reaction chamber), leaving behind seminal plasma and other cell types. The metabolic activity of stallion spermatozoa largely depends on their mitochondria and the use of oxidative phosphorylation [15]. To assess this activity, the reaction chamber contains WST-1, a water-soluble tetrazolium salt (2-(4-iodophenyl)-3-(4-nitrophenyl)-5-(2,4-disulfophenyl)-2H-tetrazolium monosodium salt). This salt is commonly used in biological assays to measure cell metabolic activity. WST-1 is cell impermeable, unlike other tetrazolium salts; therefore, its reduction is largely considered to occur at the cell surface via trans-plasma membrane electron transport [16]. Although the use of such a device for diagnostic potential is novel and ambitious, the rationale is that these highly motile sperm have the greatest odds of ascending the mare’s reproductive tract and initiating fertilization, and these are the cells on which our analysis should be based. The Samson™ isolation chamber has previously been used to diagnose the quality of thoroughbred stallion dismount semen samples [17], which, unlike standardbreds, are restricted to natural breeding. This study demonstrated that using the Samson™ isolation system allowed researchers to predict the outcome of breeding (*n* = 143; 141 mares bred by a cohort of 7 stallions) with an overall accuracy of 94.6% [17]. The primary aim of this study was to perform a detailed analysis of whole ejaculates from standardbred stallions to determine the optimal criteria to accurately predict conception and an early pregnancy result. Since the samples included in this study were meant to be cooled and transported to the location of the mares, it is typical for the stud farm, where the stallion’s semen sample is collected, to have limited information about the receiving mare except for the data available on their online pedigree. There is also an absence of any knowledge on the time of insemination or any veterinary care that the mare is provided before or after insemination. Therefore, given that previous studies suggest sperm movement is significantly associated with fertility [18,19,20], we primarily focused this study on this attribute of semen quality.

## 2. Materials and Methods

### 2.1. Materials

SpermSafe (E)™ (Breed diagnostics, Newcastle, Australia) and EquiPlus semen extender (Minitube, Ballarat, VIC, Australia) were utilized when specified during the course of this study. WST-1 was obtained from Sapphire bioscience (Redfern, NSW, Australia). Equipment and media were kept at temperatures ranging from 22–25 °C throughout semen collection and processing.

### 2.2. Preparation, and Chilling of Spermatozoa

Institutional ethical approval was secured for this project (A-2021-139), in which whole equine ejaculates (*n* = 46) were donated from 7 commercial standardbred stallions standing at stud in Wagga Wagga, Australia. Ejaculates were collected by artificial vagina (Missouri; Minitube, Ballarat, Australia) once per day, alternating daily between late morning (am) and early afternoon (pm), over a 5-week period. After collection, semen was transferred into a 50 mL Falcon tube, where volume and insemination doses were recorded. Insemination doses were determined by the stud manager dependent on the sample’s motility and volume. A concentration of 50 × 10^6^/mL total motile sperm and a total volume of 20 mL was deemed optimal for each insemination dose [21]. An aliquot of the raw ejaculate was taken to perform initial sperm isolation using the Samson™ system before the remaining ejaculate was diluted 2:1 (*v*:*v*) with Equiplus semen extender and placed into a Styrofoam box. Initial raw ejaculates were not extended before isolation to not provide the sperm cells with any substrates that could affect their motility parameters and, therefore, skew results. This box included an ice brick and a Styrofoam partition, eliminating direct contact between the semen and ice brick. The semen was transported to subsequent standardbred stud farms under these conditions (estimated 5 °C for 24 h). For the purpose of this study, a 2–3 mL aliquot of extended semen was kept on farm, under the same conditions as the transported semen, for subsequent sperm isolation using the Samson™ system.

### 2.3. Sperm Analysis

Sperm motility and concentration were quantitatively determined using iSperm^®^ Equine [22], a mobile computer-assisted sperm analyzer (mCASA; Aidmics Biotechnology Co., Ltd., Taipei City, Taiwan). The iSperm^®^ equine software (version EQ 5.5.16) and instrumentation included an iPad Mini (Apple Inc., Cupertino, CA, USA) and the proprietary microscope camera, which was set up according to the specifications of the iSperm^®^ instruction manual. iSperm^®^ motility settings, sample collector preparation, and sperm analysis were performed as described by [17]. In conjunction with total and progressive motility, the following kinematic parameters were analyzed: average path velocity [VAP], straight-line velocity [VSL], curvilinear velocity [VCL], straightness [STR], and linearity [LIN]. If the samples were outside optimal concentration parameters to return an accurate reading on the iSperm^®^ device, especially for progressive motility measurements (<100 million cells/mL required to return a progressive motility reading; app version EQ 5.5.16), a small aliquot was extended with EquiPlus semen extender to achieve a concentration suitable for iSperm analysis.

### 2.4. Sperm Isolation in the Samson™ System

The Samson™ system (Memphasys, Sydney, Australia) comprises a reaction chamber and a sample chamber divided by a 5 µm polycarbonate filter (Figure 1). As briefly described by Aitken et al. [17], 0.5 mL of SpermSafe (E)™ media (Breed Diagnostics, NSW, Australia) containing WST-1 at a concentration of 0.5 mg/mL was placed into the reaction chamber, and 0.5 mL of the ejaculate was placed into the sample chamber. The device remained at an ambient temperature of 22–25 °C for 15 min. During this time, highly motile spermatozoa passed from the sample chamber into the reaction chamber, where spermatozoa were able to reduce WST-1. Subsequently, 250 µL of sample from the reaction chamber was removed and placed into a 1.5 mL Eppendorf tube, of which 100 µL aliquots were dispensed into a 96-well place. Immediately, absorbance was read at 450 nm on a Spectrostar nano (BMG Labtech, Mornington, Australia). A media blank control containing SpermSafe (E)™ and WST-1 was used. Concurrently, the sample was analyzed via the iSperm^®^ system. This analysis was performed on the initial sample immediately after collection (pre-chill) and on the sample after it had been chilled for 24 h and rewarmed to 37 °C (post-chill).

### 2.5. Animal Information Collection

At the conclusion of the breeding season, mare and stallion information was recorded based on data available from www.harness.org.au accessed on 15 November 2021, as well as the early pregnancy results (14 days post ovulation; rectal sonograph examination).

### 2.6. Statistical Analysis

The data collected in this study were analyzed using JMP Pro16 software (SAS Institute Inc., 2004, Cary, NC, USA), and a variety of statistical methods were employed, including contingency table analysis and Pearson and Likelihood ratio Chi-square tests for mare status and stallion performance analysis. ANOVA tests for stallion and mare ages were conducted to analyze individual stallions’ reproductive success and linear regression for correlating sperm total motile count and WST-1 reduction. Additionally, stepwise discriminant analysis and Receiver Operating Curve (ROC) determinations were used to predict pregnancy. The predictive power of a variable provided by the ROC analysis was assessed by the area under the curve (AUC), which ranges from 0.5 to 1, with 0.5 indicating no difference between 2 possible outcomes and 1 indicating the explanatory variable can distinguish between 2 possible outcomes in 100% of cases. The combined predictive value of a set of continuous variables was determined by the discriminant function analysis. An accuracy of 50% would indicate the random selection, while an accuracy of 100% indicates that the selected variables can perfectly distinguish infertile from fertile ejaculates. Only eight variables were drawn into the discriminant function equation for this analysis. The data are presented as mean ± SEM, and statistical significance was considered at *p* < 0.05.

## 3. Results

### 3.1. Fertility

This analysis was based on 46 artificial inseminations, using chilled, transported semen, and involved a group of 7 stallions. Of the 46 inseminations, 22 (47.8%) resulted in an early positive pregnancy test via rectal sonograph, performed 14 days post ovulation.

### 3.2. Age

Stallion ages ranged from 12–23 years (mean 16.1 ± 1.5 years; *n* = 7). Stallion age was not correlated with the success of pregnancy. A total of 41 mares, ranging in age from 5–22 years (mean 13 ± 4.7 years) were artificially inseminated within the period of this study (due to early initial negative pregnancy tests, 5 mares required re-insemination). Once again, there was no significant difference between the ages of the mares that returned a positive pregnancy result (13.4 ± 0.9 years; *n* = 22) and those that did not (13.3 ± 1.1 years; *n* = 24).

### 3.3. Mare Status

Of the 46 inseminations, 29 were ‘dry’ (mares with no foal at foot from the previous breeding season), 7 were ‘wet’ (mare being bred at first estrus before she has weaned her foal), and 10 were maiden (mares who had never previously foaled). Mare status did not correlate with the occurrence of pregnancy within this documented study period (*p* > 0.05).

### 3.4. Stallion Performance

Pregnancy rates examined over the restricted study period differed between stallions, but this difference was found to be not significant (*p* > 0.05). Additionally, no correlation was found between the per stallion conception rate observed during this limited study period and the conception rate over the whole season (Table 1; R^2^ = 0.006).

### 3.5. Semen Quality

Representative samples were collected from all ejaculates that were used for artificial inseminations during this study. Whole ejaculates generated an average volume of 38.2 ± 2.8 mL, containing 266.7 ± 22.8 × 10^6^ sperm/mL was 73.2 ± 1.7% motile and 48.9 ± 1.8% progressively motile.

### 3.6. Pre-Chilled Ejaculate Quality

In an initial attempt to replicate the selection process of spermatozoa as they ascend the mare’s reproductive tract, the ability of the spermatozoa from the collected ejaculates to pass through a polycarbonate separation membrane with 5 µm pores was evaluated using disposable chambers (Samson™ chambers, Memphasys Ltd., Sydney, Australia). After an incubation period of 15 min, a mean of 30.7 ± 3.8 × 10^6^ sperm/mL was isolated, which exhibited significantly lower total and progressive motility compared to the parent population (Figure 2A,B). However, the fundamental velocity parameters (VAP, VSL, and VCL) did not show any significant changes (Figure 2C, D, and E, respectively). These isolated cells also exhibited less linearity in their overall movement compared to the spermatozoa collected in the whole ejaculate, showing significantly reduced levels of LIN and STR (Figure 2F and G; *p* < 0.01 and *p* < 0.001, respectively).

Supplementary to this sperm migration challenge, the ability of this isolated population of cells to reduce the tetrazolium salt WST-1 was analyzed as a further indicator of equine sperm function [17]. WST-1 is a water-soluble, membrane-impermeant tetrazolium salt, which equine spermatozoa are able to reduce into a formazan product that can be easily detected spectrophotometrically. Unlike other cell types, WST-1 in the presence of equine spermatozoa does not appear to need an intermediate electron acceptor to facilitate its reduction and instead is directly reduced at the cell surface [16,17,23]. Therefore, it was hypothesized that WST-1 reduction would not only determine sperm concentration but might also reveal the fundamental biochemical status of these cells, as other tetrazolium salts such as MTT (3-(4,5-dimethylthiazol-2-yl)-2,5-diphenyltetrazolium bromide) has performed before [24]. As predicted, WST-1 reduction within this isolated sperm population generated highly significant correlations (*p* < 0.001; R^2^ = 0.69) between WST-1 reduction and motile sperm count (Figure 3A).

### 3.7. Post-Chilled Ejaculate Quality

After the ejaculate had been diluted with EquiPlus semen extender, chilled for 24 h, and rewarmed in a 37 °C incubator for 15 min, the sperm was passed through the Samson™ system [17]. After a 15 min incubation period, a mean of 13.4 ± 1.5 × 10^6^ spermatozoa/mL was isolated and displayed significantly higher total and progressive motility than the parent population (Figure 4A,B). These isolated cells also displayed significantly higher fundamental velocity parameters, VAP, VSL, and VCL (Figure 4C–E; *p* < 0.001, *p* < 0.01 and *p* < 0.001, respectively). However, there was no significant difference in the linearity of these cells’ movement (Figure 4F,G) compared to the spermatozoa in the whole, chilled ejaculate.

Moreover, WST-1 reduction within this isolated sperm population, once again, generated highly significant correlations (*p* < 0.001; R^2^ = 0.4) between WST-1 reduction and motile sperm count (Figure 3B).

### 3.8. Prediction of Pregnancy

The semen samples were analyzed using linear discriminant function analysis to determine if any of the measured parameters could accurately predict the likelihood of pregnancy from specific insemination. Additionally, ROC analysis was conducted to confirm the predictive value of the selected criteria for semen quality.

The results of the discriminant function analysis suggest that specific parameters of semen quality, such as sperm concentration and movement characteristics, can be used to predict the likelihood of a given insemination resulting in a pregnancy. The accuracy of this prediction was improved when additional data on WST-1 reduction in Samson™ isolated cells were included in the analysis. In the pre-chilled samples, the discriminant function included STR, VSL, VAP, sperm concentration, VCL, LIN, total motility, and progressive motility and achieved an overall accuracy of 71.7% and an AUC value of 0.75 in the ROC analysis. In the post-chilled samples, the discriminant function included stallion age, VAP, LIN, STR, VCL, sperm concentration, progressive motility, and ejaculate volume and achieved an overall accuracy of 69.6% and an AUC value of 0.83 in the ROC analysis. Supplementation of these analyses with the WST-1 reduction data on Samson™ isolated cells, somewhat increased the prediction accuracy for the pre-chilled samples (73.1%) and for the post-chilled samples (71.1%). However, the AUC in the ROC analyses was slightly reduced for the pre-chilled samples (0.77) and the post-chilled samples (0.82).

A discriminant analysis was performed using all continuous variables, and stepwise analysis selected the first eight variables to be drawn into the regression equation for pre-chilled samples. These variables were STR in semen, VAP post-Samson™, VSL in semen, VAP in semen, motile sperm count post-Samson™, WST-1 absorbance, ejaculation volume, and progressive motility in semen. The full discriminant function equation is provided in Appendix A (Figure A1, Figure A2, Figure A3, Figure A4 and Figure A5). However, this equation inaccurately classified 11 inseminations, resulting in a successful prediction rate of 76.1%. The ROC analysis revealed an AUC of 0.86 for both positive and negative fertility predictions (Figure 5A).

If this analysis was repeated with the post-chilled samples, the first eight variables incorporated into the regression equation were as follows: stallion age, VCL post-Samson™, VSL post-Samson™, VSL in the semen, STR post-Samson™, motile sperm count post-Samson™, LIN in the semen, and VCL in the semen. The full discriminant function equation is provided in Appendix A (Figure A6, Figure A7, Figure A8, Figure A9 and Figure A10). The discriminant analysis equation based on the post-chilled samples incorrectly predicted the outcome of 14 inseminations, resulting in a successful prediction rate of 69.6%. Meanwhile, the ROC analysis for both positive and negative prediction of fertility showed an AUC value of 0.83 (Figure 6A).

The individual stallions involved in this study did show a range of differences regarding reproductive success over the breeding season, albeit not significant; we still felt that the stallions should be analyzed individually for a more accurate fertility prediction. Therefore, discriminant analyses were repeated for each stallion. These results are presented in Table 2 (pre-chilled data) and Table 3 (post-chilled data). The presented tables display the criteria used in the stepwise linear discriminate analyses for predicting successful pregnancy resulting from insemination. Furthermore, they indicate the accuracy of the discrimination process and the AUC values obtained from the ROC analyses. Each stallion employed a unique combination of variables to enhance the predictive process. However, collectively, these variables evaluated the quality of the ejaculate based on its volume, concentration of sperm, motility characteristics, and the quality of the sperm population after migration through the Samson™ cartridges. In addition, stallion and mare age was another criterion that was routinely drawn into the discriminate analysis. On the contrary, WST-1 reduction was not a common inclusion, and it appeared to only be utilized by 2 stallions (1 and 3) in the pre-chilled samples and 1 stallion (stallion 3) within the post-chilled samples. It is presumed that most of the information that this criterion carries is paralleled by measurements of sperm concentration and movement characteristics. We were not able to assess stallions 5–7 individually due to low *n*-values (*n* = 2, 2, and 1, respectively).

By applying these straightforward criteria, we achieved a substantial level of precision in predicting the outcome of artificial inseminations (Table 2 and Table 3). The ROC analysis also revealed an estimated overall AUC of 0.92, illustrated in Figure 5B–E for samples analyzed pre-chill, and an approximate overall AUC of 0.98 in the ROC analysis, illustrated in Figure 6B–E for samples analyzed post-chill. For stallions 1–4, our analysis provided enough information to predict the outcome of artificial inseminations with 80%, 100%, 100%, and 71.4% accuracy, respectively (average of 87.9%; 42/46 artificial inseminations), for samples analyzed pre-chilling. This value was able to be significantly increased if samples were analyzed post-chilling, with accuracy levels of 80%, 100%, 100%, and 100%, respectively (average of 95%; 44/46 artificial inseminations).

## 4. Discussion

This study is among the first to demonstrate that the quality of the stallion’s semen is one of the most important factors in determining the probability that artificial insemination will result in a pregnancy. These findings reveal that a small volume of semen (0.5 mL) may be rapidly analyzed, allowing equine breeders who utilize assisted reproductive technologies (ART) to determine whether insemination is expected to result in a pregnancy. If the likelihood of pregnancy is low, the mare may be re-inseminated during the same estrus cycle to ensure pregnancy.

Although a high level of predictive accuracy was achieved in this study, it would not have been attainable through conventional semen analysis methods. It is probable that the predictive accuracy may have been improved if a detailed sperm morphology analysis was performed alongside conventional methods [25]. However, these morphological assessments are laborious assays which require skilled technicians, and for the purpose of this study, we aimed to develop a rapid assessment that may be used in the field by unspecialized personnel. Additionally, to enhance the precision of our analysis, we could have incorporated further mare details, such as follicle management, ovulation induction, and the duration between semen collection and insemination. Unfortunately, this information is usually inaccessible from the stud farms housing the stallions. Hence, we confined our examination to publicly accessible data (mare age and progeny) obtainable from the harness racing NSW website. Using semen quality alone (age of the stallion and sperm concentration and movement characteristics), we obtained a predictive accuracy of 71.7% if semen was analyzed before it was chilled (pre-chilled). Likewise, we obtained a predictive accuracy of 69.6% if the semen was analyzed after it had been chilled for 24 h (post-chilled). Although this is undoubtedly better than the no discrimination 50% value, 70% predictive accuracy is not high enough to be significantly beneficial to the industry.

To improve diagnostic significance, conventional semen assessments were complemented with thorough analyses of isolated spermatozoa using the Samson™ chamber. The Samson™ system evaluates the capacity of equine spermatozoa to traverse a separation membrane consisting of 5 µm pores within 15 min. In the pre-chilled samples, the isolated population displayed lower levels of total and progressive motility than the parent population. However, the isolated population did manage to display decreased values for STR and LIN. This modified movement pattern suggests a reduction in the straightness and linearity of sperm progression, which is synonymous with the initiation of a more hyperactivated form of movement in this subpopulation of cells [26,27]. Previous studies have indicated that the iSperm system is less accurate than CASA when analyzing sperm velocity parameters [28]. Whilst computer-assisted sperm analysis (CASA) analysis is considered the gold standard, it is impractical to think stud farms would consider buying this technology due to the high cost. iSperm devices are more cost effective and would be better adopted within these farms. Regarding the post-chilled samples, the isolated population now displayed increased levels of total and progressive motility compared to the parent population. This discrepancy between the pre- and post-chilled samples may be explained by the high viscosity and concentration of spermatozoa in the raw ejaculate (266.7 ± 22.8 × 10^6^ sperm/mL), compared to the chilled sample, which had been extended with EquiPlus semen extender, and therefore exhibited a lower viscosity and sperm concentration (72.3 ± 6.7 × 10^6^ sperm/mL). This high viscosity and/or concentration may have impeded the ability of the spermatozoa to swim across the membrane in an efficient manner [29]. For all future studies, the authors advise dilution of ejaculates to a concentration of 100 × 10^6^ sperm/mL or less before samples are added to the Samson™ isolation system. The isolated cells obtained using the Samson™ isolation system also demonstrated the ability to directly reduce the membrane-impermeant tetrazolium salt, WST-1, without the need for an intermediate electron acceptor that is usually required in such reactions [23,30]. The oxidation of NADH to NAD+ through a plasma membrane electron transport chain likely releases electrons that enable the reduction of WST-1 at the cell surface (Figure 7; [23]). This enzymatic system may help maintain intracellular NADH/NAD+ redox balance and regulate glycolytic flux [31]. Reduction of WST-1, consistent with other tetrazolium salts appears to be highly dependent on cell viability and concentration [32]; therefore, we suspected it may be an interesting method of assessing motile sperm concentration. This proved to be the case, with cells assessed pre-chilled returning an R2 value of 0.69 and cells assessed post-chilled cells returning an R2 value of 0.4.

Combining the raw ejaculate semen quality data with information on the WST-1 reducing capacity of spermatozoa isolated through the Samson™ system resulted in a marginal improvement in the prediction of fertility outcomes. The accuracy rate increased from 71.7% to 73.1% for pre-chilled samples and from 69.6% to 71.1% for post-chilled samples. If we similarly incorporated the data from the cells post Samson™ isolation (sperm concentration and movement characteristics) and mare age, the accuracy for the pre-chilled samples again only marginally improved to 76.1% and was reduced marginally to 69.6% for the post-chill sample.

Due to variations in fertility among the stallions in the study, we optimized the semen quality parameters used for predicting fertility on a per-stallion basis. This approach resulted in a significant improvement in the accuracy of our semen analyses, with a successful classification rate of 87.9% (pre-chilled samples) and 95% (post-chilled samples) for each individual stallion. The criteria used to achieve these success rates are outlined in Table 2 and Table 3, respectively, and the discriminate function equations are summarized in Appendix A (Figure A1, Figure A2, Figure A3, Figure A4, Figure A5, Figure A6, Figure A7, Figure A8, Figure A9 and Figure A10). The ROC analyses validated the high predictive value of the chosen variables, generating average AUC values of 0.92 (pre-chilled) and 0.98 (post-chilled) for individual stallions 1–4 (individual stallions 5–7 excluded due to low *n*-values).

Overall, the predictive factors used within these individualized analyses involved traditional sperm parameters, such as ejaculate volume, motility characteristics (total motility, progressive motility, VAP, VSL, VCL, STR, and LIN), and sperm concentration, as well as novel sperm measurements performed after sperm isolation in the Samson™ chamber. Sperm parameters after Samson™ isolation were a common inclusion into the discriminant function equation for all stallions, pre- and post-chill, reinforcing that this additional isolation step is a significant one if you wish to accurately determine the outcome of any given insemination. Additionally, WST-1 reduction was drawn into the discriminant function equation for one stallion (3) within the pre-chilled samples, and for two stallions (1 and 3) within the post-chilled samples. This is presumably due to the capability of this probe to reveal some major components of sperm metabolism [31]. Furthermore, a major part of this study was to determine the optimum time to analyze the semen, pre- or post-chilling, to predict pregnancy most accurately. From our results, there does not appear to be much difference between either analyses, and analysis pre- or post-chilling will still give a result reflective of the fertilizing capabilities of the spermatozoa.

## 5. Conclusions

In conclusion, this is one of the first studies to demonstrate that semen quality, determined through rapid field analysis, is one of the most important factors in predicting the outcome of artificial insemination. The findings of this study indicate that the factors influencing stallion fertility vary among different individuals, making it difficult to develop a universal algorithm for predicting fertility. However, by customizing the analysis to individual stallions, it is possible to formulate discriminant function equations that reflect the fertilization capacity of a particular ejaculate and accurately predict the probability of a successful pregnancy after insemination. Implementing such an approach would be highly beneficial for the equine ART industry, aiding in the management of their stallions’ reproductive performance.

## Figures and Tables

**Figure 1 animals-13-01203-f001:**
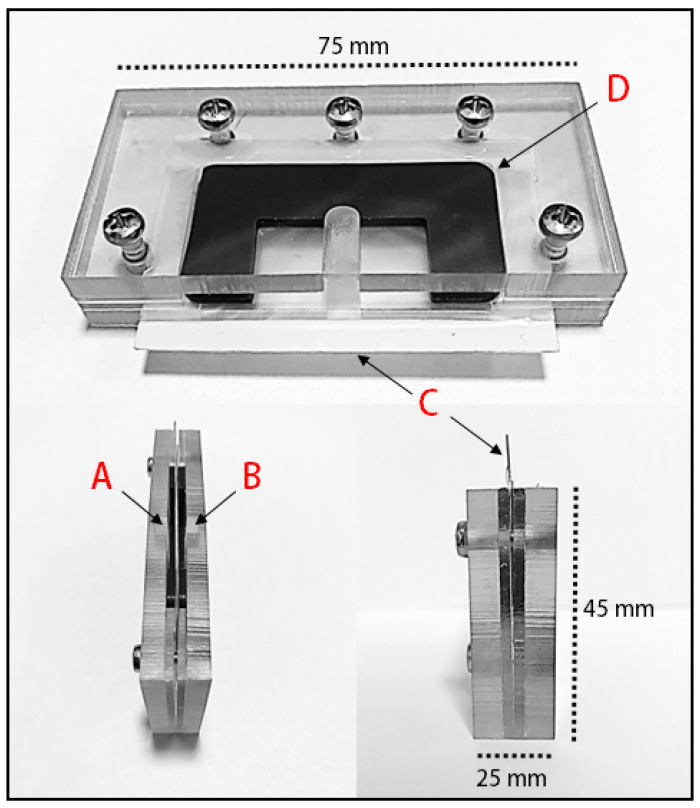
Sperm isolation in the Samson™ system consists of (A) a sample chamber and (B) a reaction chamber encased in an acrylic frame, separated by (C) a 5 µm polycarbonate filter and (D) silicone seal. 0.5 mL of SpermSafe (E)™ and 0.5 mg/mL WST-1 were deposited into the reaction chamber, followed by 0.5 mL of semen deposited into the sample chamber. The cartridge was kept at ambient temperature for 15 min, after which 250 µL was removed from the harvest chamber for analysis.

**Figure 2 animals-13-01203-f002:**
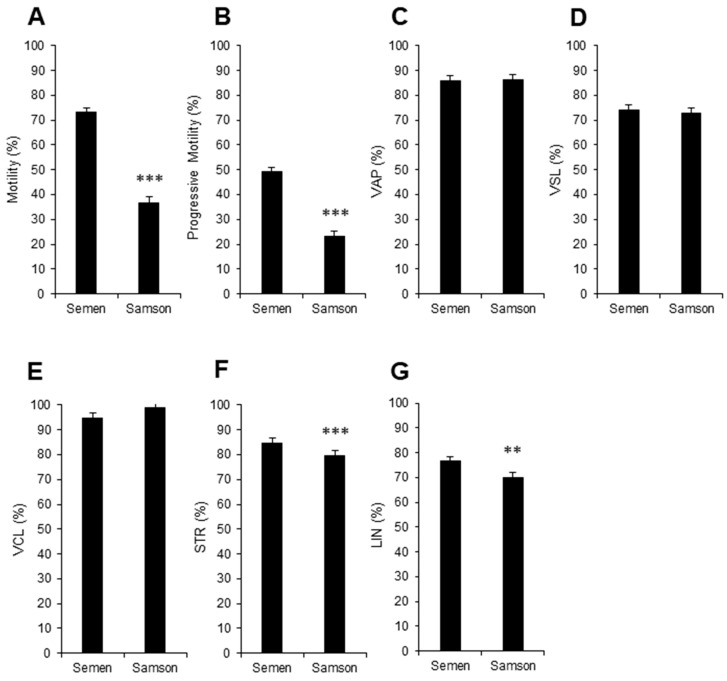
Isolation of stallion spermatozoa in a Samson™ chamber prior to chilling. (**A**) Total motility, (**B**) progressive motility, and sperm velocity parameters (**C**) VAP, (**D**) VSL, (**E**) VCL, (**F**) STR, and (**G**) LIN values of isolated samples (Samson) were compared with motility parameters from the parent population (Semen; ** *p* < 0.01; *** *p* < 0.001).

**Figure 3 animals-13-01203-f003:**
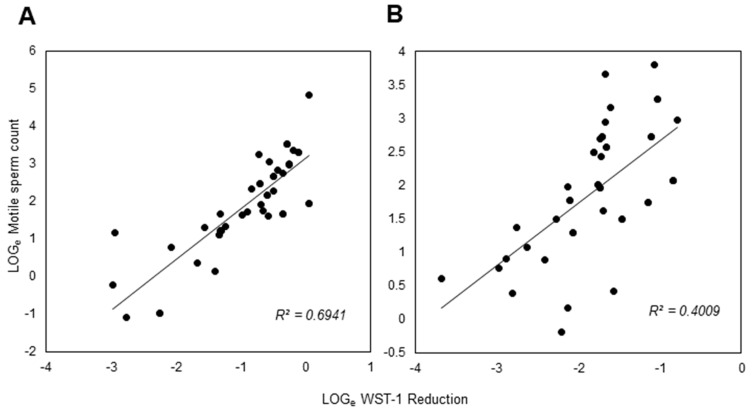
WST-1 reduction by stallion spermatozoa isolated in a Samson™ chamber. WST-1 reduction by cells isolated both (**A**) prior to chilling and (**B**) post-chilling were significantly correlated with the total number of motile cells (*p* < 0.001).

**Figure 4 animals-13-01203-f004:**
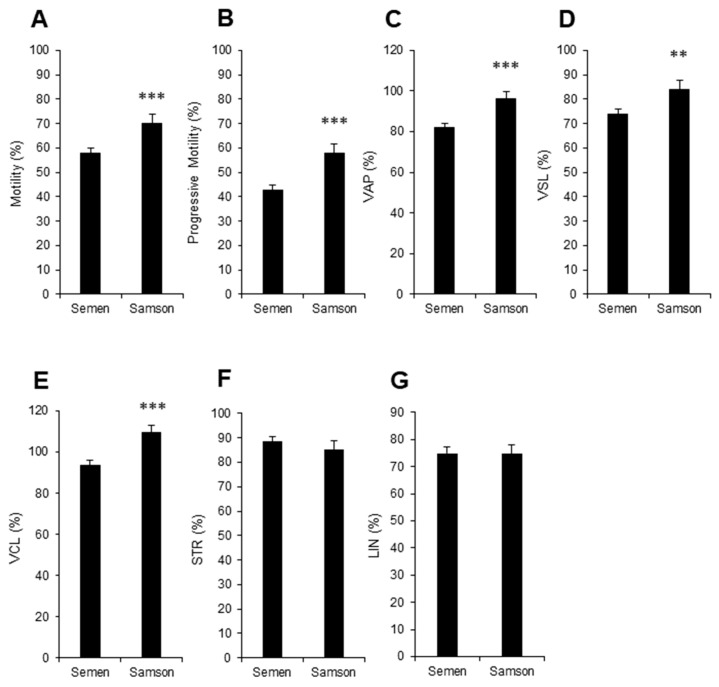
Isolation of stallion spermatozoa in a Samson™ chamber post-chilling. (**A**) Total motility, (**B**) progressive motility, and sperm velocity parameters (**C**) VAP, (**D**) VSL, (**E**) VCL, (**F**) STR, and (**G**) LIN values of isolated samples (Samson) were compared with motility parameters from the parent population (Semen; ** *p* < 0.01; *** *p* < 0.001).

**Figure 5 animals-13-01203-f005:**
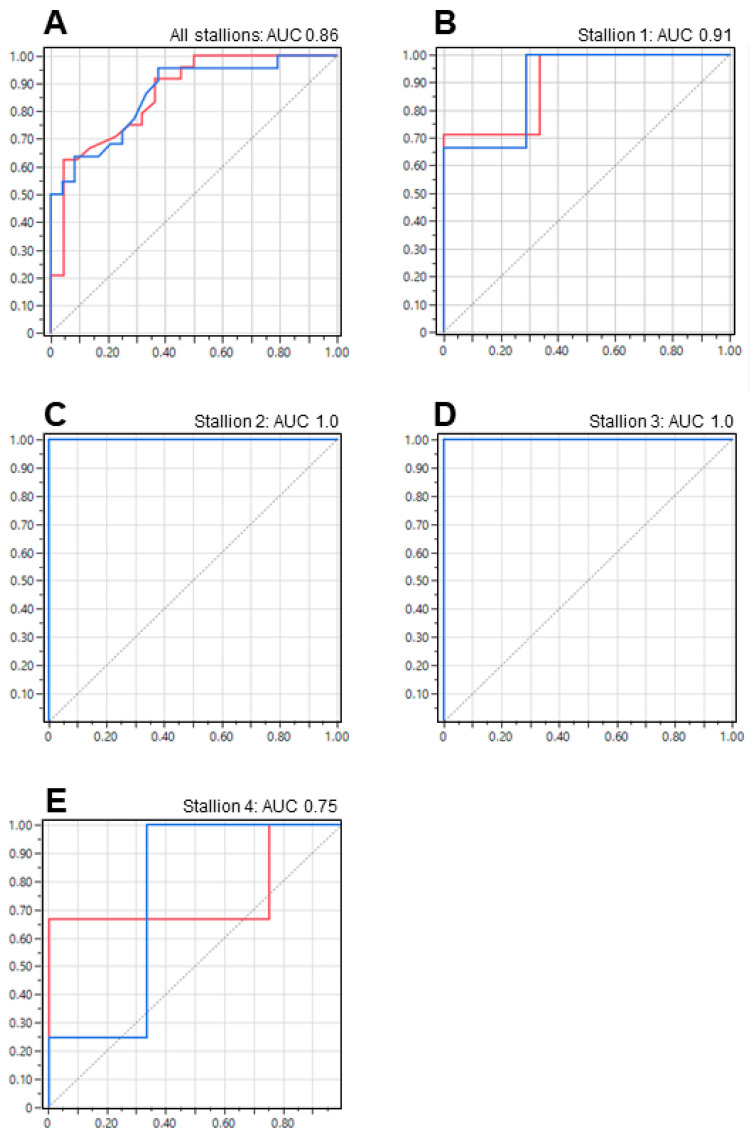
Receiver operating characteristic (ROC) curves were used to predict pregnancy based on stallion spermatozoa analyzed before chilling (pre-chill), with the aim of achieving an Area Under the Curve (AUC) value as close to 1.0 as possible. (**A**) Inclusion of all stallions resulted in a high AUC value of 0.86. When the analysis was optimized for individual stallions. AUC values in excess of 0.91 were routinely obtained (**B**–**D**), except for 1 case (**E**) which returned an AUC of 0.75.

**Figure 6 animals-13-01203-f006:**
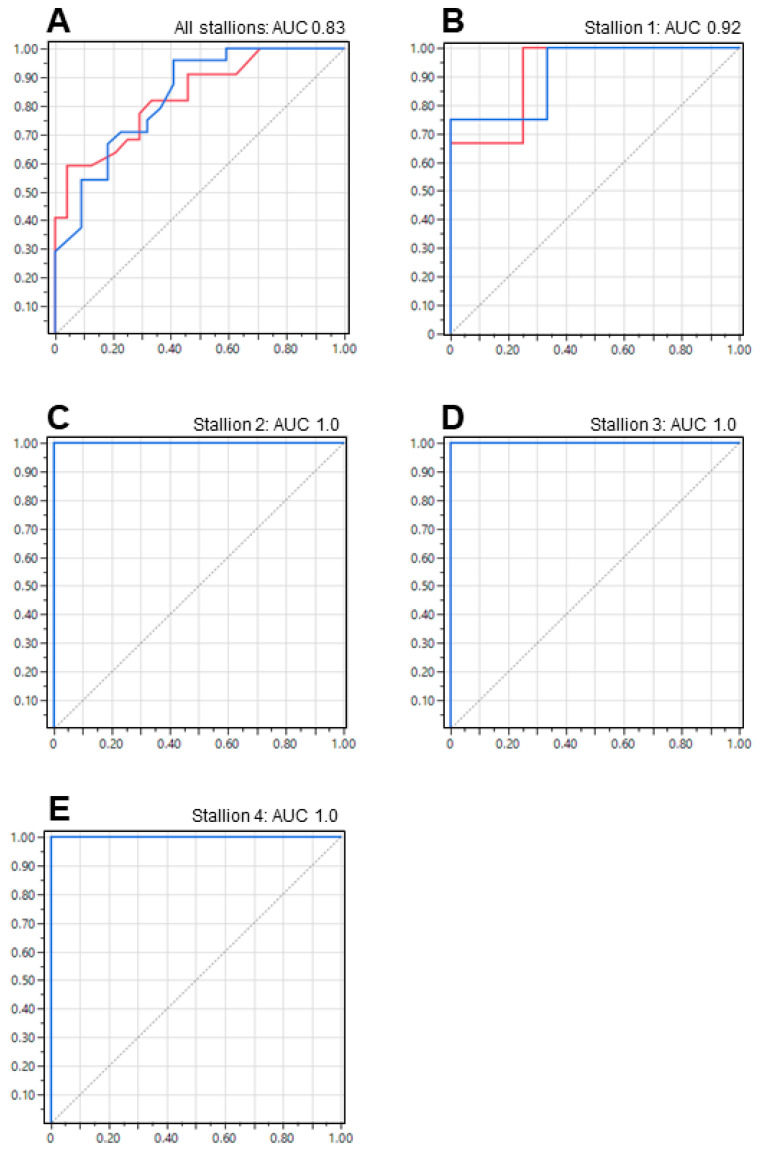
Receiver operating characteristic (ROC) curves were used to predict pregnancy based on stallion spermatozoa analyzed after chilling (post-chill), with the aim of achieving an Area Under the Curve (AUC) value as close to 1.0 as possible. (**A**) Prediction of fertility by inclusion of all stallions generated a high AUC value of 0.83. This result was only limited to individual variation of fertility between stallions. If the analyses were optimized to individual stallions, then AUC values in excess of 0.92 (**B**–**E**) were routinely obtained, suggesting a very high level of predictive accuracy.

**Figure 7 animals-13-01203-f007:**
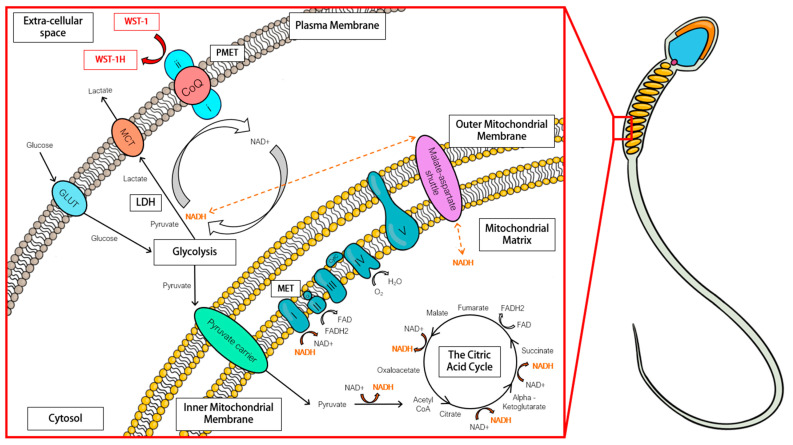
Cellular energy production, NADH recycling, and WST-1 reduction pathways in the spermatozoon of the stallion. The production of NADH in spermatozoa involves several processes, including glycolysis, the oxidation of pyruvate to acetyl-CoA, and the citric acid cycle. The mitochondria are responsible for generating 80% of the total NADH produced (10 NADH molecules per glucose molecule). NADH is re-oxidized to NAD+ through various pathways, including mitochondrial electron transport (MET), plasma membrane electron transport (PMET), and lactate dehydrogenase (LDH) activity. MET involves respiratory complexes I through IV, coenzyme Q10 (CoQ; Q), and cytochrome C (CytC), while PMET involves an inner-facing NADH oxidoreductase, (i) CoQ, and an outer-facing surface oxidase (ii). The reduction of the water-soluble, cell-impermeable tetrazolium salt WST-1 to WST-1H can be used to measure PMET activity. GLUT: Glucose transporter, MCT: monocarboxylate transporter, WST-1: 2- (4-iodophenyl) -3- (4-nitrophenyl) -5- (2,4-disulfophenyl) -2H- tetrazolium monosodium salt.

**Table 1 animals-13-01203-t001:** Pregnancy rates achieved by individual stallions.

Stallion #	Total Covers	Number Pregnant †	Number Non-Pregnant	Fertility Rate over Study Period (%)	Fertility Rate for Entire Season (%)
1	14	6	8	43	75.8
2	10	6	4	60	67
3	10	3	7	30	80.5
4	7	4	3	57	91.6
5	2	1	1	50	80.9
6	2	1	1	50	61.5
7	1	1	0	100	80

† At 14 days post ovulation. # Stallion identifier number.

**Table 2 animals-13-01203-t002:** Criteria employed to predict the success of insemination by stallion (analysis performed on semen prior to chilling).

Stallion ID	Criteria Employed	Accuracy of Prediction (*n*)	ROC Analysis AUC
1	STR following migration Ejaculate concentration VCL in the semen Mare age VCL following migration * VSL following migration Stallion age	80% (10)	0.91
2	Total motility following migration Motile sperm count post migration Ejaculate concentration Ejaculate volume LIN following migration Progressive motility in the semen LIN in the semen Mare age	100% (14)	1.0
3	Total motility in the semen Concentration following migration WST-1 absorbance Ejaculate concentration Motile sperm count following migration Mare age	100% (10)	1.0
4	Concentration following migration Mare age LIN following migration Progressive motility following migration VAP following migration	71.4% (7)	0.75

* Migration in the Samson™ chamber.

**Table 3 animals-13-01203-t003:** Criteria employed to predict the success of insemination by stallion (analysis performed on semen post-chilling).

Stallion ID	Criteria Employed	Accuracy of Prediction (*n*)	ROC Analysis AUC
1	VCL in the semen Total motility in the semen Stallion age Mare age WST-1 absorbance Total motility following migration * Progressive motility following migration	80% (10)	0.92
2	LIN following migration Stallion age STR following migration VCL in the semen Total motility following migration Mare age LIN in the semen Progressive motility following migration	100% (14)	1.0
3	WST-1 absorbance VCL in the semen STR following migration VAP following migration Total motility in the semen	100% (10)	1.0
4	VSL following migration Mare age LIN following migration VSL in the semen Ejaculate concentration	100% (7)	1.0

* Migration in the Samson™ chamber.

## Data Availability

Data reported in this study, may be accessed from the Harvard Dataverse or via the link https://doi.org/10.7910/DVN/9PMNTX accessed on 10 of October 2022.

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
