# Peer review of "Predicting the Outcome of Equine Artificial Inseminations Using Chilled Semen"

_animals, 2023, doi:10.3390/ani13071203_

Round 1

Reviewer 1 Report

General comments:

The main objective of the manuscript titled “Predicting the outcome of equine artificial inseminations using chilled semen” is to establish if analysis of a stallion ejaculate can be used to accurately predict the success rate of an artificial insemination.

The main limitations of this study is the small sample size used to predict fertility outcome, and that the mare-factor is not included in the analysis. 

There is no information regarding the mares that were inseminated. In the results it appears that mare age and history was recorded and used in the statistical analyses. Management of the mares is very important and is highly correlated to the pregnancy outcome. E.g. follicle control, induction of ovulation, post AI ultrasound to check for ovulation, fluid? Post AI-treatment, how many times were the mares inseminated in each cycle, etc. Where the mares at the same farm and bred/examined by the same vet? Was the semen used at the day of arrival (24 hours after collection/processing)?

The rationale behind the use of the Samson isolation chamber is not clear, and needs to be included in the introduction including relevant literature supporting its use.

There are some deficiencies that need to be addressed before considering its publication.

There are some criteria that need to be fulfilled to get valid results from the iSperm. One of them is dilution of the semen samples before analysis of progressive motility. If dilution was not performed, the results from the progressive motility of the pre-chilled samples should be interpreted with caution.

The iSperm might not be the most accurate semen analyzer for this study. One study demonstrated low levels of agreement for velocity measurements (when compared to CASA system), and that the concentration range should be in the range 30 million/ml to approximately 240 millions/ml to obtain accurate results (Moraes et al., 2019). These results should at least have been included (and discussed) in the discussion.

 Detailed comments:

Introduction:

The isolation chamber needs to be introduced here, including relevant literature.

Line 62-64: Belongs to Materials and Methods section

Line 68-76: Belongs to results and discussion

Material and Methods:

Line 92: how was insemination doses recorded after the transfer of the semen to a 50 ml Falcon tube?

Line 105: App version of iSperm?

Was the raw semen and the semen harvested from the Samson chamber  pre-chill diluted before iSperm analysis? According to other studies, dilution before analysis for e.g. progressive motility is essential to obtain valid results.

Line 147: Information regarding the mares, gynecological examinations, treatment pre- or post AI, number of insemination per cycle, management etc. is missing

Statistical analysis:

It is not clear which tests/methods that were used for the different analyses. At least a multivariate regression model should be used for evaluation of different independent variables effect on the outcome of the dependent variable (e.g. pregnancy outcome).

Where ejaculate number considered a variable in the statistical analysis?

Results:

Line 204-207: References to the statement that membrane in the Samson chamber mimics the spermatozoa’s migration through the mares reproductive tract is needed.

Line 229-231: Move to introduction

Figure 3: does not fit into the results section

Line 272: Prediction of pregnancy:

The sample size (N=7 stallions, who bred 7-14 mares) for predicting pregnancy is very limited and with the exclusion of all mare parameters/risk factors (except from mare age) from the analysis makes these results very questionable.

Discussion:

Line 362-364: to determine the probability that an artificial insemination will result in pregnancy will need a much larger sample size, when the “mare factor” is not an integrated part of the analysis.

Reviewer 2 Report

The manuscript entitled “Predicting the outcome of equine artificial inseminations using chilled semen.” aimed to establish if analysis of a stallion ejaculate may be used to accurately predict if artificial insemination would result in pregnancy. I have identified some major issues that should improve the quality of the manuscript:

1-      Please write references according to the journal's instructions.

2-      I think lines 62-68 are the aim of the study please clarify that.

3-      Please delete lines 68-76 from the introduction because these are results.

4-      Please write the fertility trial in detail in materials and methods.

5-      Please revise figures 3 and four and their footnote.

6-      Please don’t mention again all results, figures, and tables in the discussion part, summarize the important results and discussed them in depth.

Round 2

Reviewer 2 Report

The authors revised the manuscript according to the referee's suggestions. The paper is now suitable for publication.

Author Response

The authors would like to thank reviewer 2 for taking the time to review the submitted changes to the manuscript.